# Multiple and Consecutive Genome Editing Using i-GONAD and Breeding Enrichment Facilitates the Production of Genetically Modified Mice

**DOI:** 10.3390/cells12091343

**Published:** 2023-05-08

**Authors:** Carolina R. Melo-Silva, Cory J. Knudson, Lingjuan Tang, Samita Kafle, Lauren E. Springer, Jihae Choi, Christopher M. Snyder, Yajing Wang, Sangwon V. Kim, Luis J. Sigal

**Affiliations:** 1Department of Microbiology and Immunology, Thomas Jefferson University, Philadelphia, PA 19107, USA; 2Department of Emergency Medicine, Thomas Jefferson University, Philadelphia, PA 19107, USA

**Keywords:** i-GONAD, CRISPR, genetically modified mice

## Abstract

Genetically modified (GM) mice are essential tools in biomedical research. Traditional methods for generating GM mice are expensive and require specialized personnel and equipment. The use of clustered regularly interspaced short palindromic repeats (CRISPR) coupled with improved-Genome editing via Oviductal Nucleic Acids Delivery (i-GONAD) has highly increased the feasibility of producing GM mice in research laboratories. However, genetic modification in inbred mouse strains of interest such as C57BL/6 (B6) is still challenging because of their low fertility and embryo fragility. We have successfully generated multiple novel GM mouse strains in the B6 background while attempting to optimize i-GONAD. We found that i-GONAD reduced the litter size in superovulated pregnant females but did not impact pregnancy rates. Natural mating or low-hormone dose did not increase the low fertility rate observed in superovulated B6 females. However, diet enrichment had a positive effect on pregnancy success. We also optimized breeding conditions to increase the survival of small litters by co-housing i-GONAD-treated pregnant B6 females with synchronized pregnant FVB/NJ companion mothers. Thus, GM mice generation was increased by an enriched diet and shared pup rearing with highly fertile females such as FVB/NJ. In the present study, we generated 16 GM mice using a CRISPR/Cas system to target individual and multiple loci simultaneously or consecutively. We also compared homology-directed repair efficiency using different methods for LoxP insertion for conditional knockout mouse production. We found that a two-step serial LoxP insertion, in which each LoxP sequence was inserted individually in different i-GONAD procedures, was a low-risk high-efficiency method for generating floxed mice.

## 1. Introduction

Genetically modified (GM) mouse strains are essential tools to understand the function and importance of genes in mammals and as models of human disease. Since the first transgenic mouse strain was generated [1,2], the tools and methods to produce novel GM mice have greatly improved and an enormous number of GM mouse models have been made. The discovery of the bacterial genetic editing system CRISPR (clustered regularly interspaced short palindromic repeats) and its adaptation into a technology to edit mammalian genomes allowed the direct genome editing of single-cell embryos [3] and has almost completely replaced the use of homologous recombination in embryo stem (ES) cells for targeted genome modification [1,2] and of random mutagenesis for phenotype-driven mouse models [4].

In vitro embryo genome editing using CRISPR greatly facilitated GM mouse creation. However, embryo collection, in vitro handling, microinjection or electroporation, reintroduction into new females, and retrieval of viable pups pose challenges to most research laboratories because these steps require highly trained personnel and expensive equipment. These methods also use large numbers of mice. The improved-Genome editing via Oviductal Nucleic Acids Delivery (i-GONAD) [5,6], which bypasses the need for in vitro embryo handling, delivers CRISPR reagents directly into the pregnant females’ oviducts. This method makes it feasible for research laboratories or departments to make their own GM mice.

Much of the mouse biomedical research uses the C57BL/6 (B6) mouse strain. Thus, most GM mouse strains are generally needed in the B6 background. Unfortunately, compared to other mouse inbred strains, B6 mice have poor reproductive fitness [7] and embryo survival [8]. Thus, genetic modification is usually performed in other mouse strains, followed by multiple backcrosses to B6 mice. This process is lengthy, the degree of completeness is difficult to assess, and closely linked genes from the original strain usually remain, possibly compromising the results. Therefore, optimizing the conditions for direct genetic modification of B6 mice is of utmost interest for biomedical research. Here, we compared conditions that could impact pregnancy rates, litter size, and pup survival while successfully generating multiple GM mouse strains using i-GONAD in B6 mice. We observed a consistent negative effect of i-GONAD on embryo survival. Most importantly, we found that a diet with higher fat and protein content increased pregnancy rate, and companion FVB/NJ mothers helped with pup rearing, increasing the survival of i-GONAD pups. Thus, special attention to breeding conditions can maximize success in generating B6 GM mice.

## 2. Material and Methods

### 2.1. Mice

All females used for i-GONAD were 8–12 weeks old, except when we used experienced breeders, which were 4- to 6-months old. Males used as studs for i-GONAD were 3- to 6-months old. C57BL/6NCrl (B6), Crl:CD1 (ICR) or FVB/NCrl (FVB/NJ) mice were purchased from Charles River (Wilmington, MA, USA) directly for experiments or as breeders. C57BL/6J (B6) and C57BL/6-Tg(UBC-GFP)30Scha/J (UBC-GFP) mice were purchased from the Jackson Laboratory (Bar Harbor, ME, USA). All GM mouse strains generated in the present work were derived directly from B6 breeders purchased from Charles River or from offspring of B6 breeders purchased from Charles River, except for *Gpr15^DREmut1^* and *Gpr15*^DREΔ^ mice (Table 1), which were derived from mice purchased from the Jackson Laboratory. Colonies were bred at Thomas Jefferson University under specific pathogen-free conditions. Mice were group housed on individually ventilated cages and fed ad libitum on 5010 LabDiet (Quakertown, PA, USA), unless specified otherwise. All mice procedures were carried out according to the Eighth Edition of the Guide for the Care and Use of Laboratory Animals of the National Research Council of the National Academies. All experiments were approved by Thomas Jefferson University’s Institutional Animal Care and Use Committee under protocol number 02065, “Operation of the TJU Transgenic Mouse Facility”.

### 2.2. CRISPR Reagents and Design

Cas9 protein conjugated with a nuclear localization signal was purchased from PNA Bio, (Cat no CP02, Thousand Oaks, CA, USA). Primers, single-stranded DNA (ssDNA—up to 200 bases), trans-acting CRISPR RNA (tracrRNA) and CRISPR RNA (crRNA) reagents were purchased from IDT Integrated DNA Technologies (Coralville, IA, USA). Long ssDNAs (>200 bases) were purchased from GenScript (Piscataway, NJ, USA) for *Adipor1*^fl/fl^ or prepared in-house for *Gpr15*^DREmut1^ and *Gpr15*^DREΔ^ (Van et al., unpublished) and *Mov10*^Δexon12–14^ mice by PCR of the genetic locus with specific primers, followed by treatment with T7 exonuclease from New England Biolab (Ipswich, MA, USA) or by Guide-it Long ssDNA Production System v2 from Takara (cat no 632666, San Jose, CA, USA). crRNAs were designed using the CRISPOR web tool [9] and are listed on Sup Table 1. Primers were designed using the NCBI’s Primer-BLAST online tool [10] or by CRISPOR web tool. crRNA and primer design were based on NCBI’s Genome Reference Consortium Mouse Build 39 (GRCm39) as reference for B6 mice. Repair templates were designed to target the same DNA strand as the crRNA and contained homology arms of 60–80 bp. For RNP preparation, 30 μM crRNA:tracrRNA complex, 40 μM ssDNA (up to 200 bases), 0.8–1 μg/μL ssDNA (>200 bases), 1 mg/mL Cas9 protein, and 0.02% Fast Green FCF (cat no F7252, Sigma-Aldrich, St. Louis, MI, USA) were mixed. For experiments in which 2 genomic targets were independently and concomitantly edited, both crRNA:tracrRNA and both repair templates were mixed in an individual concentration of 30 μM and 40 μM, respectively.

### 2.3. Hormone Treatment and Natural Mating

For superovulation, B6 females were superovulated by inoculating 5 IU PMS (Prospec cat no hor-272, Mt Vernon, NY, USA) intraperitoneally (IP) at 12:00 p.m. two days before mating and 5 IU of human chorionic gonadotropin (HCG, Sigma-Aldrich Cat no C1063) at 12:00 p.m. right before mating. i-GONAD was performed the day after mating from 4:00 p.m. to 6:00 p.m. For low-dose PMS-only treatment, the schedule was similar, but the females were inoculated with 1.2 IU PMS/10 g of body weight, and the HCG was skipped. For natural mating, B6 females were co-housed in large cages (15 females/cage) and exposed to dirty male bedding 3 times a week for two weeks. One day before i-GONAD, vaginal smears from B6 females were checked under the microscope for cytological evaluation of estrous cycle stages [11]. Females under proestrus and estrus were mated. Pregnancy was confirmed by observation of body and behavior changes characteristic of pregnant females, such as belly and nipple enlargement and nesting behavior.

### 2.4. i-GONAD

i-GONAD was performed according to Gurumurthy et al. [5] with minor modifications. Briefly, 5–15 B6 females were superovulated, PMS-only treated, or checked for estrus or diestrus stage by vaginal smear analysis and, subsequently, mated 1/1 with B6 males at noon on the day before i-GONAD. Plugs were checked in the evening and again early in the morning. At 4 p.m.–6 p.m., plugged females were anesthetized with 3.5% isoflurane in an anesthesia chamber. After anesthesia was achieved, the mice were transferred to a warm pad, and anesthesia was maintained using a nose cone. The mice were shaved and cleaned, treated with 0.01 mg carprofen (Zoetis, cat no 141–199, Parsippany-Troy Hills, NJ, USA) to reduce pain, and with eye lubricant to prevent eye dryness. The oviducts and ovaries were exposed by small dorsal incisions and rested on PBS-soaked paper wipes. The RNP cocktail was injected into the oviducts using a custom-made needle (Hamilton cat no 7803-04-32GA/12 mm/20°, Parsippany-Troy Hills, NJ, USA) attached to a 2.5 μL glass syringe (Hamilton, cat no 7632-01). The oviducts were covered with PBS-soaked wipes and electroporated with constant 100 mA current using a CUY21EDITII (BEX, Tokyo, Japan), as previously described [5]. The oviducts and ovaries were returned to the abdominal cavity, and the dorsal incision was closed with surgical clips. Females were treated post-surgery with 0.04 mg bupivacaine (Auromedics, cat 55150-167-10, East Windsor, NJ, USA) and placed in a warm cage to recover from the anesthesia. I-GONAD-treated females were checked for pregnancy two weeks after the procedure, and pregnant females were housed together for litter delivery.

### 2.5. Food Enrichment, Fostering, and Co-Rearing

5010 or 5K52 rodent diets (LabDiet) were given to females for at least two weeks before i-GONAD. For fostering, foster B6 or ICR females were superovulated one day before those subjected to i-GONAD and mated or naturally mated. At 19–20 days post-fertilization, pregnant females subjected to i-GONAD were sacrificed by cervical dislocation. A dorsal incision was made on the skin, and the whole skin was removed by pulling downwards from the incision. The internal organs were exposed by an abdominal incision with scissors. The uterus was removed and placed on a warm pad covered with clean paper. Newborns were quickly released from the uterus. Amniotic fluid was removed from the nose with a clean tissue paper, and abdominal massage was applied to improve breathing.

Fostering adults were removed from the fostering cages, some of their pups were euthanized to maintain the litter size, and the i-GONAD newborns were gently rubbed with bedding and feces from the fostering cage to improve their chances of adoption and mixed with the remaining pups in the fostering cages. Fostering adults were quickly returned to the fostering cage. Successful fostering was recorded a few days later. For co-rearing, FVB/NJ females were either superovulated or treated with a low dose of PMS simultaneously with the i-GONAD B6 females. Two pregnant i-GONAD B6 and one pregnant FVB/NJ females were co-housed in static microisolated rat cages (18 × 6 × 9 inches). The number of pups born and those that survived were recorded.

### 2.6. Genetic Characterization of Founders and Data Analysis

i-GONAD founders were genotyped by PCR and Sanger sequencing. Sequencing results were analyzed with ApE, A plasmid Editor [12], and ICE Analysis-Synthego web tool (https://ice.synthego.com/). Linked off-targets or non-linked off-targets with a cutting frequency determination higher than 20% were selected for sequencing analysis. Exon deletion was assessed by Reverse Transcription and quantitative PCR (qPCR) of total RNA extracted from spleens of mice, as previously described [13]. Data were analyzed with Prism 8 Software. Unpaired Student’s *t*-test with Welsh’s correction and two-sided Fisher’s exact test were used to determine *p* values. * *p* value < 0.05, ** *p* value < 0.01, *** *p* value < 0.001 and **** *p* value < 0.0001.

## 3. Results

### 3.1. i-GONAD Reduces Litter Size

To set up i-GONAD, we superovulated B6 females and mated them with B6 male mice homozygous for a GFP transgene under the Ubiquitin C promoter (UBC-GFP mice). I-GONAD was performed at 0.7 days post-fertilization, as described by others [5], but the RNP cocktail with crRNA specific for GFP [14] and without repair ssDNA was only injected into the left oviduct, while the right oviduct was left untreated as a negative control (Figure 1A). At 18 days post-fertilization, fetuses were harvested. All fetuses from the untreated oviduct were GFP^+^, whereas most fetuses from the treated oviduct were GFP^−^. Thus, i-GONAD successfully knocked out the GFP gene with high efficiency by non-homologous end joining (NHEJ).

Despite the high frequency of gene-specific editing, the number of fetuses in the i-GONAD-treated oviduct was significantly lower than in the untreated oviduct (Figure 1B). The treated but not the untreated oviducts also contained some malformed fetuses, indicating that the CRISPR reagents or the electroporation but not the surgery affected pup viability.

Superovulation has been shown to consistently increase oocyte and litter numbers [15,16]. However, the average litter size of B6 females under standard colony breeding conditions was larger than in superovulated B6 females subjected to i-GONAD on both oviducts (Figure 1C), confirming an effect of i-GONAD on embryo or pup survival in the B6 strain.

Considering the B6 low-to-medium reproduction fitness [7], we next compared the effect of i-GONAD on litter size in FVB/NJ females, which are known for their excellent reproduction fitness [17]. We found that the litter size of superovulated and i-GONAD-treated FVB/NJ was significantly smaller than the litters of superovulated FVB/NJ females that did not undergo i-GONAD (Figure 1D). Together, our observations showed that i-GONAD negatively impacted the litter size in mouse strains with low or high reproductive fitness.

### 3.2. Food Enrichment, However, Not Natural Mating, Increases Pregnancy Rate in B6 Females

We also compared pregnancy rates in superovulated B6 females subjected or not to i-GONAD (Figure 2A). We found that i-GONAD did not affect pregnancy rate in superovulated B6 young females. However, the average pregnancy rate of 20% observed in our superovulated females was low and a pitfall for genome editing. Thus, we aimed to optimize husbandry conditions to increase pregnancy success.

We first tested whether diet could improve pregnancy rates. We compared the life-cycle nutrition standard diet 5010 LabDiet (29% protein, 13% fat, and 58% carbohydrate) to the primary breeding diet 5K52 LabDiet (22% protein, 16% fat content, and 62% carbohydrate). We observed that 5K52 increased the pregnancy rate from 20 to 40% (Figure 2B). Mice were henceforth given 5K52 diet for at least two weeks before i-GONAD.

We observed that i-GONAD-treated B6 females often cannibalized their pups, especially when they had small litters. B6 females are known to sometimes cannibalize their first litter [18]. Thus, we tested whether experienced B6 female breeders, which had given birth and fully weaned their first litter with no cannibalization, could be better subjects for i-GONAD. However, we found that the pregnancy rate in experienced breeders was significantly lower than in young females (8 to 10 weeks) (Figure 2C). Therefore, young females were used in subsequent i-GONAD experiments.

It has been shown that superovulation can cause delivery complications in some mouse strains [16]. Thus, natural mating or hormone treatment with a single low dose of pregnant mare serum (PMS) is often preferred for i-GONAD [19]. We compared pregnancy rates in i-GONAD-treated B6 females that were superovulated, naturally mated, or treated only with a low dose of PMS two days before mating. We found that natural mating and low-dose PMS did not improve pregnancy rates in i-GONAD-treated B6 strains compared to superovulation (Figure 2D). Overall, our data showed that young superovulated females fed with a high-protein, high-fat diet were the most suitable subjects to maximize pregnancy rate in B6 mice subjected to i-GONAD.

### 3.3. Co-Housing with Synchronized Good Mothers but Not Non-Survival Cesarean and Fostering Increases the Survival of i-GONAD B6 Pups

Next, we tested whether foster mothers could reduce cannibalization and increase the survival of i-GONAD pups. For that, we replaced some of the just-delivered pups of superovulation-synchronized B6 or ICR mothers, with the pups obtained by non-survival cesarean delivery at 20 days post-fertilization from i-GONAD-treated B6 females. However, adoption success varied greatly among experiments (Figure 3A), resulting in the loss of many i-GONAD pups.

Next, we co-housed i-GONAD-treated pregnant B6 females with superovulation-synchronized pregnant FVB/NJ females. Cohousing started a week before delivery and lasted until weaning. We found that the presence of FVB/NJ companion females, which are vigorous breeders [17], increased the number of live i-GONAD B6 pups compared to i-GONAD-treated B6 females that were co-housed only with other i-GONAD-treated B6 females (Figure 3B). FVB/NJ females delivered their pups 1–2 days before the B6 females and assisted the feeding and grooming of the newborn i-GONAD B6 pups. This effect had a greater impact on the survival of small litters (≤5), which were cannibalized more frequently than larger litters (Figure 3C). The chances of small litter survival, defined here as ≥50% of pups surviving until weaning, increased from 23% to 90% when the i-GONAD B6 moms were co-housed with companion FVB/NJ moms. Hence, B6 litter survival could be increased by cohousing i-GONAD-treated B6 pregnant females with a companion female with high reproductive fitness, which would help nursing and caring for newborn i-GONAD-derived B6 pups.

### 3.4. Using i-GONAD to Produce Gene Knockouts, Point Mutations, and Promoter Modifications

Genome editing using inbred strains such as B6 mice requires extra attention to husbandry conditions, but the high efficiency of i-GONAD turned the cumbersome, labor-intensive task of generating GM mouse strains into a relatively simple low hands-on laboratory procedure that could easily be established by most labs with mouse breeding experience. Within a few years and with non-exclusively dedicated staff, we have established and used i-GONAD to generate ten knockout strains, one single nucleotide polymorphism (SNP) knock-in strain, three promoter deletion strains, and two floxed strains (Table 1).

To generate knockout mice, we used single-stranded DNA (ssDNA) of up to 200 bases as repair templates. These contained stop codons and small exogenous sequences.

We simultaneously edited highly homologous genes such as the type I interferon (IFN-I) locus on chromosome 4 (Figure 4A). We used one ssDNA repair template and one crRNA targeting identical sequences in *Ifna1* and *Ifna5.* We obtained mice with perfect HDR insertion in *Ifna1* and *Ifna5*. Additionally, *Ifna7*, with a predicted 70% crRNA cutting efficiency, was modified by a non-sense mutation after NHEJ. Thus, we obtained triple knockout mice for *Ifna1*, *Ifna5,* and *Ifna7* (*Ifna1,5,7*^−/−^ mouse, Figure 4B). The small exogenous sequences (20 bp) were useful during breeding for distinguishing heterozygous, homozygous, and wild-type mice by agarose gel electrophoresis (Figure 4B). This experiment indicated that i-GONAD permitted successfully targeting homologous sites in a single experiment with only one crRNA and ssDNA construct, and HDR could occur at more than one site when using a single crRNA and ssDNA.

We also attempted to use CRISPR double strand break-driven HDR to modify in a single i-GONAD experiment two non-homologous genes, each targeted with their own ssDNA and crRNA. Non-homologous crRNA and ssDNA repair templates encoding stop codons with specificity for either *Ifnb1* or *Ifna4* were mixed and used for i-GONAD. Out of a single i-GONAD procedure, we retrieved three founders. The first founder had a single nucleotide insertion, resulting in a frameshift and early stop codon in the *Ifnb1* gene without *Ifna4* modification (*Ifnb1*^−/−^ mouse, Figure 4C). A second founder had a nucleotide insertion in *Ifnb1,* resulting in a frameshift and early stop, and an imperfect HDR-mediated event, in which part of the repair-template-encoded exogenous sequence was inserted and a nine-nucleotide sequence was deleted in *Ifna4* (*Ifna4,b1*^−/−^ mouse, Figure 4D). Another mouse had insertion/deletions (indels) in *Ifnb1* and *Ifna4*, with a large deletion surrounding the *Ifna4* cutting site. Thus, i-GONAD successfully targeted two non-homologous sites, delivering single- and double-knockout mice in the same experiment. However, we did not obtain any mouse with perfect HDR in the two genes when these genes were targeted in a single experiment. Still, we obtained perfect HDR insertion of stop codons in the *Ifna4* gene when only the *Ifna4*-specific crRNA and ssDNA were used in a different i-GONAD procedure (*Ifna4*^−/−^ mouse, Figure 4E). The same strategy was used to generate *Rgs1*^−/−^ and *Rgs10*^−/−^ mice (Table 1).

We also used single short ssDNA repair templates to delete the NF-κB binding site (Positive regulatory domain II-PRDII) [20] in the *Ifnb1* gene promoter (*Ifnb1*^ΔPRDII^ mouse, Figure 4F) and to produce a point mutation in the *Elovl2* gene (*Elovl2*^C234W^ mouse, Table 1). Unfortunately, the *Elovl2*^C234W^ founders failed to produce viable homozygous offspring on the B6 background, differing from previous reports [21]. Interestingly, heterozygous males and females were produced from a female homozygous founder but not from a male homozygous founder (Table 2). Moreover, subsequent breeding of heterozygous males to either B6 wildtype females or heterozygous females did not result in pups. Literature suggests this point mutation may interfere with spermatogenesis [22], and backcross to the 129/SV mouse strain should increase the chances of producing viable male offspring [23]. Thus, the failure of the *Elovl2*^C234W^ founders to produce homozygous offspring likely results from male infertility rather than the use of i-GONAD.

A long ssDNA repair construct and two-crRNAs were used to mutate the aryl hydrocarbon receptor (AHR) binding sites in the *Gpr15* locus. Perfect HDR did not occur, and the planned founder was not obtained. However, from this experiment, we obtained two useful strains: (a) one strain with desired mutations in two AHR binding sites and a 129 bp deletion upstream (*Gpr15^DREmut1^* mouse, Table 1) produced by imperfect HDR repair; and (b) one complete deletion of the locus around two AHR binding sites (*Gpr15*^DREΔ^ mouse, Table 1) produced by NHEJ driven by the two cutting events (Van et al., unpublished).

### 3.5. Using i-GONAD to Produce Conditional Knockout Mice

We tried three strategies to insert two LoxP sites (flox) for conditional knockout mouse production. (a) Simultaneous and independent insertion of two LoxP using two crRNAs and two ssDNA repair constructs, each containing a LoxP. With this strategy, we attempted to flox exon 3 in *Occ1*, exons 12–14 in *Mov10,* or the entire *Ifna* locus (the 14 *Ifna* genes plus *Klhl9*) by inserting one LoxP sites between *Ifnb1* and *Ifna15* and another between *Ifna1* and *Ifne*. This strategy did not produce floxed mice. However, we obtained mice with partial or full deletion of the intervening sequences, which were repaired by NHEJ or imperfect HDR. These modifications resulted in *Occ1*^Δexon3^ mice (Figure 5A), with deletion of exon 3; *Mov10*^Δexon12−14^ mice (Figure 5B), with deletion of exons 12–14; and *Ifna16-6*^−/−^ mice (Figure 5C), with deletion of the *Ifna16*, *Ifna2*, *Ifnab*, *Klhl9*, *Ifnz*, and *Ifnz*-like predicted genes, the *Ifna7*, *Ifna11,* and *Ifna6* genes, and the intergenic regions. The 160 kb genomic deletion present in *Ifna16-6*^−/−^ mice likely resulted from on-target and off-target cutting at an intergenic region between *Ifna13* and *Ifna16*. Thus, even when the original goal was not achieved, this strategy resulted in valuable GM mouse strains and suggests that two crRNAs without a repair construct could be used as a strategy to rapidly delete sequences of various lengths. Of note, we observed that the MOV10 RNA helicase activity, compromised by the partial disruption of Motif I and the absence of Motifs II and III, following the exon 12–14 deletion [24,25], is not essential for embryo and brain development, as described in *Mov10*^−/−^ mice [26,27,28], as *Mov10*^Δexon12−14^ mice breed normally and have no evident phenotype.

(b) Simultaneous linked insertion of two LoxPs using two crRNA and a long ssDNA repair construct containing the two LoxPs and the intervening sequence. We used this strategy to successfully generate *Adipor1^fl/fl^* mice with a floxed exon 2 (Table 1). While we successfully produced these mice, the mean HDR efficiency of this method was only 5% (Figure 5E).

(c) Serial insertion of two LoxPs using two crRNAs and two short ssDNA repair constructs, each containing a LoxP. In this case, the first LoxP was inserted and single-LoxP founder mice were bred to homozygosity. Homozygous mice were then used as acceptors to introduce the second LoxP. Using this strategy, we produced *Ifna*^fl/fl^ mice (the first LoxP between *Ifnb1* and *Ifna15* and the second between *Ifna1* and *Ifne*, Figure 5D). While relatively slow, serial LoxP insertion was very effective, with 56% HDR efficiency (Figure 5F).

## 4. Discussion

Since i-GONAD was first described in 2015 [6], the procedure has been adopted by several research laboratories. i-GONAD has been optimized for inbred strains such as B6 [19], and it has been used to generate GM wild mice [29]. However, we showed here that the RNP treatment or the electroporation negatively impacted embryo survival even when using the optimal constant 100 mA current for electroporation. This observation, combined with the relatively low breeding and fertility fitness of B6 mice, often compromises the creation of novel GM strains.

We found that providing food enrichment with increased fat and protein content doubled the pregnancy success of i-GONAD-treated B6 females. Our observation indicated that husbandry can be optimized for improving B6 breeding fitness and that perhaps other dietary and housing enrichment protocols, such as vitamins, hay, or nesting materials, may further increase the fertility of B6 females for i-GONAD.

In contrast to another study [19], we did not observe a positive effect of natural mating or of low-dose hormone treatment on pregnancy success. In that previous study, the authors increased the pregnancy rate to an average of 56%, similar to the pregnancy rate of superovulated females and higher than that of low-dose hormone-treated females in our experiments (Figure 2D). This difference may be a consequence of different breeding conditions, or, perhaps, the food enrichment used in our study masked the effect of natural mating or low-dose hormone treatment in the fertility of B6 females kept on a life-cycle nutrition standard diet. We also found that superovulated young females were superior for i-GONAD than superovulated experienced mothers.

Given the high frequency of pup cannibalism, we tested whether non-survival cesarean surgery followed by fostering would result in better pup survival. However, we did not observe consistent improvement. On the other hand, co-housing the i-GONAD-treated females with synchronized companion FVB/NJ moms significantly increased the survival of i-GONAD pups, especially when the litters were small. This increased survival could be due to delivery induction in the i-GONAD females often delayed in superovulated females or the nursing and grooming provided by the FVB/NJ females. Considering the impact observed in the survival of small litters, it is possible that different co-housing numbers, such as 1/5 or 2/2 FVB/NJ companion/i-GONAD B6 females, may further improve the survival of i-GONAD pups. In addition, we did not consistently measure whether i-GONAD-treated females present altered prenatal behavior, such as anxiety or reduced nesting behavior.

We simultaneously targeted two homologous sites with the same crRNA and ssDNA to modify two highly similar genes and two non-homologous sites with independent crRNAs and ssDNAs to flox genomic segments of different lengths. While we were able to modify the two homologous genes, we were unable to generate floxed mice using this approach. Our results contrasted with one study [30] but agreed with another [31]. Of note, simultaneous cutting efficiency was high for two homologous sites targeted by a single crRNA or for two non-homologous sites targeted with two independent crRNAs. This suggested that cutting efficiency may not be a limiting step to modifying multiple targets. On the other hand, we obtained mice with perfect HDR at the two sites when targeting homologous sequences with a single ssDNA but not when targeting non-homologous sequences with independent ssDNAs. This suggested that non-homologous ssDNAs may interfere with each other. In our experiments, we designed the two crRNAs to target the same DNA strand and the ssDNA constructs with the same orientation to prevent self-annealing of the LoxP sequences. It is possible that targeting different DNA strands or other modifications in the ssDNA repair template may improve HDR efficiency for the simultaneous insertion of two LoxP sites.

We also successfully used one long ssDNA with two LoxP sequences to flox one exon in a single step. However, the HDR efficiency was low, and the cost of using a long ssDNA was much higher than when using short ssDNA. Furthermore, this method was restricted to flox relatively short introns. Thus, in our experience, the consecutive insertion of two LoxP sequences is efficient, straightforward, and low risk.

## Figures and Tables

**Figure 1 cells-12-01343-f001:**
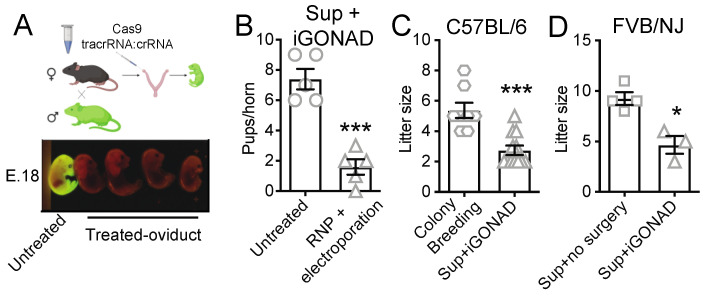
RNP injection and electroporation during i-GONAD reduces embryo survival. (**A**) Sup B6 females were mated with UBC-GFP males and, subsequently, treated with i-GONAD for GFP deletion only on the left oviduct while the right oviduct was left untreated (BioRender.com). At 18 days post-fertilization, embryos were harvested from both oviducts and GFP^+^ and GFP^−^ pups were counted. (**B**) Number of pups observed in oviducts that were untreated or that were injected with RNP and electroporated in Sup i-GONAD-treated B6 females. (**C**) Litter size from B6 colony breeders and from Sup i-GONAD-treated B6 females, in which both oviducts were injected and electroporated. (**D**) Litter size from Sup and from Sup i-GONAD-treated FVB/NJ females, in which both oviducts were injected and electroporated. Bar graphs shown mean ± SEM of independent i-GONAD procedures. * *p* value < 0.05 and *** *p* value < 0.001 (Unpaired Student’s *t*-test with Welsh’s correction). Sup: superovulated.

**Figure 2 cells-12-01343-f002:**
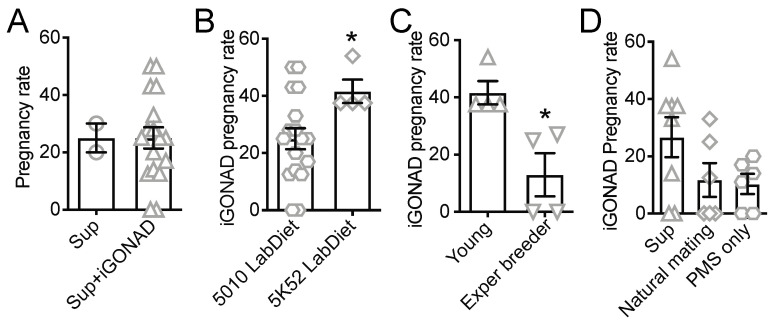
Food enrichment, but not natural mating or low hormone treatment, increase pregnancy rates for i-GONAD-treated B6 females. (**A**) Pregnancy rates observed in Sup and Sup i-GONAD-treated B6 females. (**B**) Pregnancy rates observed in Sup i-GONAD-treated B6 females fed on 5010 or 5K52 LabDiet. (**C**) Pregnancy rates observed in Sup i-GONAD-treated young (8 to 12 weeks) or Exper breeder (4 to 6 months) B6 females. (**D**) Pregnancy rates observed in i-GONAD-treated treated with either superovulation or low dose of PMS or subjected to natural mating. Bar graphs shown mean ± SEM of independent i-GONAD procedures. * *p* value < 0.05 (Unpaired Student’s *t*-test with Welsh’s correction). Sup: superovulated; Exper: experienced; PMS: pregnant mare serum.

**Figure 3 cells-12-01343-f003:**
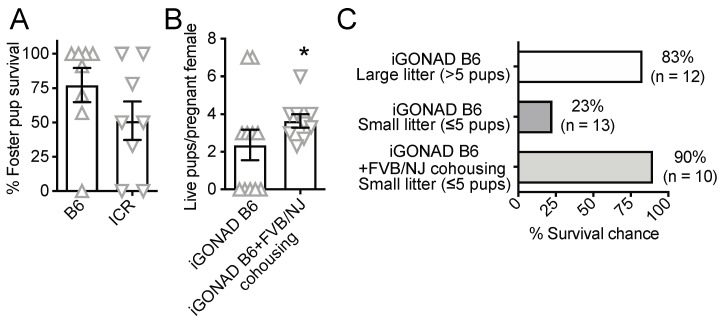
Co-housing of Sup i-GONAD-treated pregnant B6 females with synchronized FVB/NJ mothers increases the survival of small i-GONAD litters. (**A**) Percent survival of pups after cesarean section and fostering with B6 or ICR foster females. (**B**) Number of live pups per Sup i-GONAD-treated pregnant B6 female when co-housed with other Sup i-GONAD-treated pregnant B6 females or with other Sup i-GONAD-treated pregnant B6 females and pregnant FVB/NJ females. Bar graphs shown mean plus SEM of independent i-GONAD procedures. * *p* value < 0.05 (Unpaired Studen’s *t*-test with Welsh’s correction). (**C**) Percent survival chance of i-GONAD-treated large (>5 pups) and small (≤5 pups) litters when Sup i-GONAD-treated pregnant B6 females were co-housed only among themselves, or when they were co-housed among themselves and with pregnant FVB/NJ females. Litters in which at least 50% of the pups survived until weaning date were counted as surviving (n depicts number of litters tested).

**Figure 4 cells-12-01343-f004:**
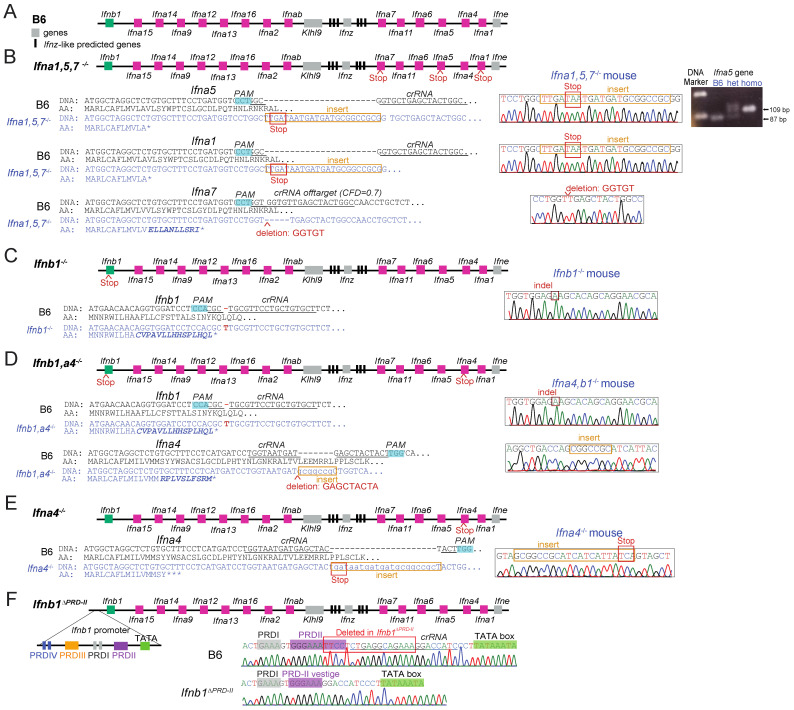
Small ssDNA repair templates are highly efficient for introducing exogenous sequences for generation of knockout GM mice. (**A**) Chromosome 4 scheme depicting the mouse IFN-I locus. (**B**) Stop codon insertions by ssDNA-driven HDR in *Ifna1* and *Ifna5* genes; and 5 nt deletion driven by NHEJ in *Ifna7* gene resulting in an early stop codon in *Ifna1,5,7*^−/−^ mouse. A total of 2% agarose gel showing nested PCR product, depicting a 20 bp insertion of multiple stop codons within the upstream coding sequence of the *Ifna5* gene in the *Ifna1,5,7*^−/−^ mice (**C**) 1 nt insertion driven by NHEJ in *Ifnb1* gene, resulting in an early stop codon in *Ifnb1*^−/−^ mouse. (**D**) 1 nt insertion driven by NHEJ in *Ifnb1* gene, resulting in an early stop codon and an imperfect HDR event, which resulted in a 7 nt insertion in *Ifna4* gene, also resulting in an early stop codon in *Ifnb1,a4*^−/−^ mouse. (**E**) Stop codon insertions by ssDNA-driven HDR in *Ifna4* gene in *Ifna4*^−/−^ mouse. (**F**) Partial deletion of the NF-kB binding site in *Ifnb1* promoter (PRDII) by NHEJ.

**Figure 5 cells-12-01343-f005:**
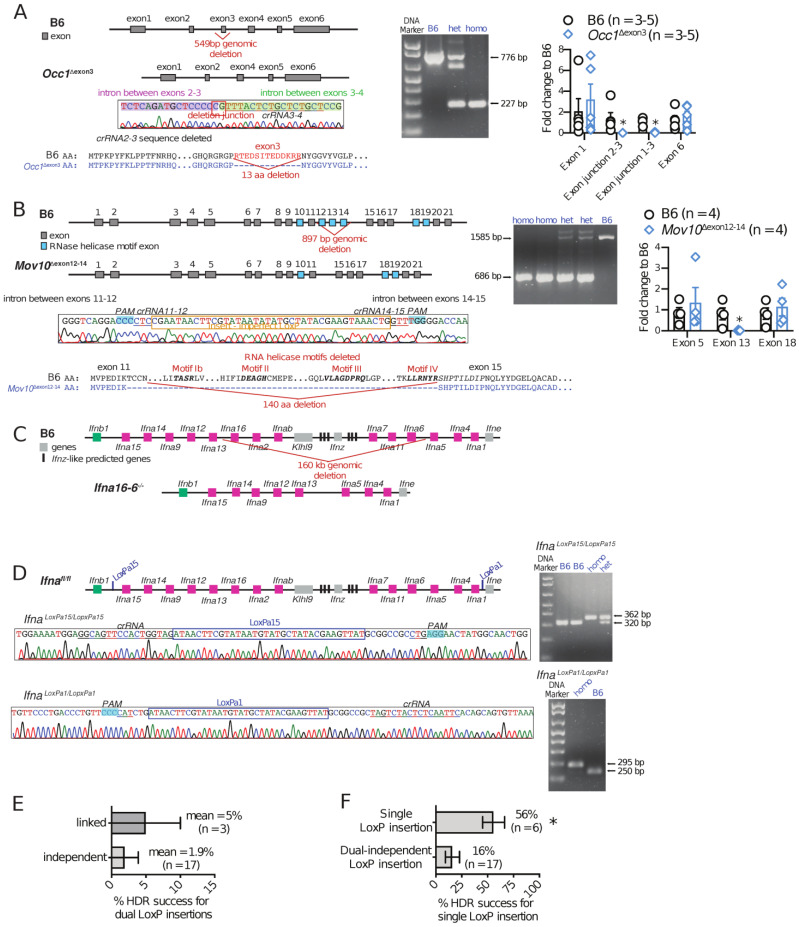
Serial LoxP insertions is an effective strategy for generation of floxed strains. (**A**) Deletion of *Occ1* gene exon 3 by NHEJ after treatment with 2 flanking crRNAs. A total of 1% agarose gel, showing PCR product depicting the 549 bp deletion, which includes exon 3 and quantitative PCR showing absence of exon 3 in *Occ1* mRNA variants in *Occ1*^Δexon3^ mice. Bar graphs show mean ± SEM of two pooled independent experiments. * *p* value < 0.05 (Unpaired Student’s *t*-test with Welsh’s correction). (**B**) Deletion of *Mov10* gene exons 12, 13, and 14 by NHEJ after treatment with 2 flanking crRNAs. A total of 1% agarose gel, showing PCR product depicting the 897 bp deletion, which includes exons 12, 13, and 14 and quantitative PCR, showing absence of exon 13 in *Mov10* mRNA in *Mov10*^Δexon12−14^ mice. Bar graphs show mean ± SEM of pooled two independent experiments. * *p* value < 0.05 (Unpaired Student’s *t*-test with Welsh’s correction). (**C**) Deletion of a 160 kb genomic region of the IFN-I locus in chromosome 4 by NHEJ after 2 on-target and 1 off-target cutting events, resulting in deletion of 6 *Ifna* genes, *Ifnz* gene, *Ifnz*-like predicted genes, and *Klhl9* gene. (**D**) 2 LoxP insertions, LoxPa15 (between *Ifnb1* and *Ifna15* genes) and LoxP1 (between *Ifna1* and *Ifne*), flanking all *Ifna* genes in chromosome 4. A total of 2% agarose gel, showing PCR product depicting a 42 bp insertion for LoxPa15 and 45 bp insertion for LoxP1 in the *Ifna*^fl/fl^ mouse. (**E**) Frequency of perfect HDR when inserting two LoxP sequences in the same i-GONAD experiment using a single >200 bp-long ssDNA repair construct containing the two LoxP sites (linked) or two ≤200 bp ssDNA repair templates containing one LoxP site each (independent). (**F**) Frequency of perfect LoxP insertion at one or two sites when attempting to insert one or two LoxP sites with only one or two short ssDNA repair templates. Bar graphs show mean ± SEM of independent i-GONAD procedures. * *p* value < 0.05 (Unpaired Student’s *t*-test with Welsh’s correction).

**Table 1 cells-12-01343-t001:** Description of all transgenic mice generated in the present work and genome editing efficiency observed using CRISPR and i-GONAD.

Transgenic Mouse	Edited Locus	Genome Modification	CRISPR Repair Template	Number of Founders	CRISPR Efficiency *
*Gpr15^DREmut1^* (Van et al., unpublished)	Chr 12, *Gpr15* gene promoter	Disruption of two AHR binding sites in *Gpr15* gene locus by multiple nucleotide substitution	long ssDNA for specific nucleotide substitutions in AHR binding site in the *Gpr15* gene promoter	5 **	20% imprecise HDR 40% total
*Gpr15*^DREΔ^ (Van et al., unpublished)	Chr 12, *Gpr15* gene promoter	Full deletion around two AHR binding sites in Gpr15 locus	long ssDNA for specific nucleotide substitutions in AHR binding site in the *Gpr15* gene promoter	5 **	40% total
*Elovl2* ^C234W^	Chr 13, *Elovl2* gene	One nucleotide mutation for cysteine to tryptophan substitution in the amino acid residue 234 of ELOVL2 protein	ssDNA for specific SNP	18	33% HDR 66% total
*Adipor1* ^fl/fl^	Chr 1, *Adipor1* gene	LoxP insertions flanking *Adipor1* gene exon 2	long ssDNA for linked dual LoxP insertions flanking target exon	10	20% HDR
*Mov10* ^Δexon12−14^	Chr 3, *Mov10* gene	Deletion of exons 12, 13, and 14 by NHEJ	long ssDNA for linked dual LoxP insertions flanking exons 12, 13, and 14	16 (2 i-GONAD procedures)	18.7% HDR on only 1 side 25% flanking sequence excision 75% total
*Occ1* ^Δexon3^	Chr 10, *Occ1* gene	Deletion of exon3 by NHEJ	2 ssDNA for one-step independent dual LoxP insertions flanking exon 3	4	50% HDR on only 1 side 50% flanking sequence excision 75% total
*Rgs10* ^−/−^	Chr 7, *Rgs10* gene	STOP codon insertion	ssDNA for STOP codon insertion in ORF	9	44% HDR 67% total
*Rgs1* ^−/−^	Chr 1, *Rgs1* gene	STOP codon insertion	ssDNA for STOP codon insertion in ORF	8	25% HDR 37.5% total
*Ifnb1* ^ΔPRDII^	Chr 4, *Ifnb1* gene promoter	PRDII/NF-κβ binding site deletion in *Ifnb1* gene promoter	ssDNA for specific nucleotide deletions in promoter	1	100% total
*Ifnb1* ^−/−^	Chr 4, *Ifnb1* gene	NHEJ indel resulting in early STOP codon	ssDNA for STOP codon insertion in ORF	3 **	33% HDR 100% total
*Ifna4,b1* ^−/−^	Chr 4, *Ifna4* and *Ifnb1* genes	STOP codon insertion in *Ifna4* gene and NHEJ indel resulting in early STOP codon in *Ifnb1* gene	ssDNA for STOP codon insertion in ORF	3 **	33% HDR 100% total
*Ifna4* ^−/−^	Chr 4, *Ifna4* gene	STOP codon insertion	ssDNA for STOP codon insertion in ORF	8	62% HDR 100% total
*Ifna1,5,7* ^−/−^	Chr 4; *Ifna1*, *Ifna5* and *Ifna7* genes	STOP codon insertion in *Ifna1* and *Ifna5* genes and NHEJ indel resulting in early STOP codon in *Ifna7* gene	ssDNA for STOP codon insertion in ORF	7	14% HDR 28% total
*Ifna6-16* ^−/−^	Chr 4, *Ifna* locus	Large chromosome deletion by NHEJ resulting in deletion of *Ifna16*, *Ifna2*, *Ifnab*, *Ifna7*, *Ifna11*, *Ifna6*, and *Klhl9* genes	2 ssDNA for one-step independent dual LoxP insertions in intergenic sequences	5	40% precise HDR only on 1 insertion site 20% imprecise HDR
*Ifna^Ifna1-Ifne^* ^fl^	Chr 4, *Ifna* locus	upstream LoxP insertion at intergenic sequence (between *Ifnb1* and *Ifna15* genes) and downstream insertion of LoxP at intergenic sequence (between *Ifna1* and *Ifne* genes)	2 ssDNA for one-step independent dual LoxP insertions in intergenic sequences	7	14% HDR only on downstream LoxP insertion
*Ifna* ^fl/fl^	Chr 4, *Ifna* locus	upstream LoxP insertion at intergenic sequence (between *Ifnb1* and *Ifna15* genes) on *Ifna^Ifna1-Ifne^* ^fl^ founder offspring	ssDNA a single LoxP insertion in intergenic sequence	9	56% HDR

* Total CRISPR-driven genetic modification (HDR + NHEJ); ** Multiple transgenic mice derived from a single i-GONAD procedure.

**Table 2 cells-12-01343-t002:** Mutant and heterozygous male *Elovl2*^C234W^ mice produced using i-GONAD failed to produce offspring on B6 background.

Breeders (Male, Female) *	Male Genotype	Female Genotype	# Number of Pups
WT B6, i-GONAD founder	*Elovl2* ^+/+^	*Elovl2* ^−/−^	40
i-GONAD founder, WT B6	* Elovl2 * ^ −/− ^	* Elovl2 * ^ +/+ ^	0
WT B6, F1 het	*Elovl2* ^+/+^	*Elovl2* ^+/−^	6
F1 het, WT B6	* Elovl2 * ^ +/− ^	* Elovl2 * ^ +/+ ^	0
WT B6, F1 het	*Elovl2* ^+/+^	*Elovl2* ^+/−^	14
WT B6, F1 het	*Elovl2* ^+/+^	*Elovl2* ^+/−^	31
WT B6, F1 het	*Elovl2* ^+/+^	*Elovl2* ^+/−^	21
F1 het, WT B6	* Elovl2 * ^ +/− ^	* Elovl2 * ^ +/+ ^	0
F1 het, WT B6	* Elovl2 * ^ +/− ^	* Elovl2 * ^ +/+ ^	0
F1 het, WT B6	* Elovl2 * ^ +/− ^	* Elovl2 * ^ +/+ ^	0
F1 het, F1 het	* Elovl2 * ^ +/− ^	* Elovl2 * ^ +/− ^	0
F1 het, F1 het	* Elovl2 * ^ +/− ^	* Elovl2 * ^ +/− ^	0

* Founders from i-GONAD or founders’ offspring (F1) were used for breeding *Elovl2*^C234W^ mice. In blue, breeding steps when mutant or heterozygous (het) males were used are indicated.

## Data Availability

Not applicable.

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
