# Peer review of "Multiple and Consecutive Genome Editing Using i-GONAD and Breeding Enrichment Facilitates the Production of Genetically Modified Mice"

_cells, 2023, doi:10.3390/cells12091343_

Round 1

Reviewer 1 Report

Genetically modified (GM) mice are one of the essential models used in biomedical research. Traditional methods used to generate genetic mouse models are expensive, complex, and time-consuming. CRISPR via i-GONAD has been increasingly applied for its ease of use to generate GM mice. However, GM in inbred mouse strains such as B6 is still challenging due to low fertility. Studies looking for methods to optimize GM using CRISPR and i-GONAD to generate GM B6 mice will maximize success and save time when creating genetic models.

Melo-Silva et al investigated the effect of i-GONAD on litter and methods of improving the success of generating GM B6 mice. Consistent with past research, authors discovered that i-GONAD reduces litter size. Natural mating did not increase the pregnancy rate while high fat and protein diet did. The authors also discovered that young B6 females have higher pregnancy rates but often cannibalized their pups. To solve this problem, authors discovered that co-housing with a synchronized mom helps with the survival of B6 pups. The authors also provided info and outcome on different types of GMs as part of the study

The paper is clearly written with visually clear figures. As GM mice are one of the most used biomedical tools and many labs have started to use i-GONAD to generate models, this research is significant to help mitigate the challenges researchers face when using i-GONAD for GM B6 mice. A few minor comments are listed below to improve the manuscript.

Minor comments:

1.       It was not very clear how pregnancy rates are defined and measured (Fig 2). Authors can clarify this in the method section or the figure legend.

2.       Add in qualitative observations if i-GONAD treated B6 females have different prenatal behaviors (e.g, pup bedding prep, anxiety, etc.)

3.       In the discussion section, the authors highlighted discrepancies from the past study (lines 406-411). It will help the readers to better comprehend the discrepancies by reviewing if the difference is the relative pregnancy rate or absolute pup size and how significant are the differences observed.

4.       Some minor formatting inconsistencies need to be fixed. For example, spacing between line 198-199 is non-consistent with other sub-section headings. Additionally, texts from line 199-209 are in bold

Author Response

Dear reviewer,

We thank you for the comments and suggested changes. They helped to refine the final manuscript. Please find below a point-by-point response to the comments.

  1. It was not very clear how pregnancy rates are defined and measured (Fig 2). Authors can clarify this in the method section or the figure legend. Response: We included a description in the method section of how we defined and measured pregnancy rates.
  2. Add in qualitative observations if i-GONAD treated B6 females have different prenatal behaviors (e.g, pup bedding prep, anxiety, etc.) Response: Unfortunately, we did not consistently measured prenatal behaviors to be able to draw conclusions on the matter. We added a note on the conclusions to highlight this point as a research perspective.
  3. In the discussion section, the authors highlighted discrepancies from the past study (lines 406-411). It will help the readers to better comprehend the discrepancies by reviewing if the difference is the relative pregnancy rate or absolute pup size and how significant are the differences observed.  Response: We included a more detailed description of the differences found in pregnancy rates between our study and the previous study. We cannot directly compare differences in absolute litter size because the previous study and our study used slightly different electroporation conditions, which directly impact embryo survival but not pregnancy rates. 
  4. Some minor formatting inconsistencies need to be fixed. For example, spacing between line 198-199 is non-consistent with other sub-section headings. Additionally, texts from line 199-209 are in bold.
  5. Response: We formatted the manuscript accordingly.

The manuscript was edited with changes tracked to highlight all new and adjusted content.

Reviewer 2 Report

This article shows that in i-GONAD, the use of companion mothers and enriched diets improves pregnancy and litter viability in the B6 strain, which is the most used strain in the biomedical research, but previous methods have resulted in low pregnancy rates and challenging. This paper is well researched and many strains have been successfully created. It fully deserves to be published, but the following points could be improved.

1) Title. Please add the title about “enrich diet”.

2) Abstract. Please revise the content to be more specific and in line with the content.

3) The correspondence between Figure 4 and Figure 5 and Table 1 is confusing, so please make it easier to understand by numbering them or unifying the order.

Author Response

Dear reviewer,

We thank you for the comments and suggested changes. They helped to refine the final manuscript. Please find below a point-by-point response to the comments.

1) Title. Please add the title about “enrich diet”. 

Response: We modified the title to "Multiple and consecutive genome editing using iGONAD and breeding enrichment facilitates the production of genetically modified mice". We believe that breeding enrichment will include both diet enrichment and companion mom environmental enrichment.

2) Abstract. Please revise the content to be more specific and in line with the content.

Response: We revised the abstract content as recommended by the reviewer. Changes in the abstract are tracked in the manuscript.

3) The correspondence between Figure 4 and Figure 5 and Table 1 is confusing, so please make it easier to understand by numbering them or unifying the order.

Response: We revised Figure 4, 5 and Table 1 correspondence for clarity purposes, as requested by the reviewer.

The manuscript was edited with track-changes to highlight all new and adjusted content.

Reviewer 3 Report

Melo-Silva and collaborators presented an interesting and well-described manuscript, that must be useful for the scientific field. i-GONAD is one of the valuable strategies for producing genetically modified animals that emerged after CRISPR technology came up. It is a young strategy that needs to be teased, and the present work has shown exactly this, with an honest description of the results. It can certainly be used as an encourager and guide for someone trying to standardize i-GONAD in the lab. They explored from the housing animals strategy, passing by breeding and going through different genome editing strategies regarding NHEJ and HDR, even using double targetings sites with single or double crRNAs. They detailed different strategies and results and were sincere about the failures. Besides, they showed a table with a list of reached animals using their strategy and providing the goals. Sometimes I missed the statistic mark on the graphics in the manuscript, where the difference statistic was unclear in the text. Also, I missed information about the abbreviation meaning of " Sup" in the figures, despite what may be understandable. These last comments may help the manuscript but don't remove the merit, which I considered a good contribution tool for the animal models development field.  

Author Response

Dear reviewer,

We thank you for the comments and suggested changes. They helped to refine the final manuscript. Please find below a point-by-point response to the comments.

"Sometimes I missed the statistic mark on the graphics in the manuscript, where the difference statistic was unclear in the text. Also, I missed information about the abbreviation meaning of " Sup" in the figures, despite what may be understandable."

Response: We added Sup meaning in the Figure Legends as to state clearly the abbreviation. We added statistics information to missing figures and fig legends.

The manuscript was edited with track-changes to highlight all new and adjusted content.